# Biopesticide Evaluation from Lab to Greenhouse Scale of Essential Oils Used against *Macrosiphum euphorbiae*

**Lana Dunan** [1,2], **Tara Malanga** [1], **Philippe Bearez** [1], **Sylvain Benhamou** [1,3,4], **Lucie S. Monticelli** [1], **Nicolas Desneux** [1], **Thomas Michel** [5,*] and **Anne-Violette Lavoir** [1,*]

1   Université Cote d'Azur, INRAE, CNRS, UMR ISA, 06000 Nice, France; dunan.lana@gmail.com (L.D.); tara@fabre-malan.ga (T.M.); Philippe.Bearez@inrae.fr (P.B.); sylvain.benhamou@hotmail.fr (S.B.); lucie.monticelli@gmail.com (L.S.M.); nicolas.desneux@inrae.fr (N.D.)
2   School of Biological Sciences, University of Aberdeen, Aberdeen AB24 3FX, UK
3   CNRS, VetAgro Sup, UMR 5558, Laboratoire de Biométrie et Biologie Evolutive, Université de Lyon, 69622 Villeurbanne, France
4   INRAE, INSA Lyon, BF2I, UMR 203, Universite de Lyon, 69621 Villeurbanne, France
5   Institut de Chimie de Nice, Université Côte d'Azur, CNRS, UMR 7272, Parc Valrose, CEDEX 2, 06108 Nice, France
*   Correspondence: thomas.michel@univ-cotedazur.fr (T.M.); anne-violette.lavoir@univ-cotedazur.fr (A.-V.L.)

**Abstract:** Aphids are recognized as a major threat to economically important crops. Their control is predominantly based on synthetic insecticides that are detrimental to human health and the environment. Botanical pesticides based on essential oils (EOs) are a promising alternative. In this study, the entomotoxicity of green anise and fennel EO fumigation was tested on the potato aphid *Macrosiphum euphorbiae*. Three different settings of increasing scale were considered (leaflet, whole plant and greenhouse) to appraise the consistency of EO impact from controlled laboratory to greenhouse production conditions. $LC_{50}$ values for green anise and fennel were 6.6 μl $L^{-1}$air and 12.2 μl $L^{-1}$air, respectively, based on dose-response curves in leaflet experiments but fennel EO induced phytotoxicity. EO efficiency was confirmed at the whole-plant scale. In the greenhouse experiment, fennel EO exhibited greater efficiency than at the laboratory scale equaling green anise EO efficiency but both EOs showed delayed phytotoxicity, illustrating the importance of long-term monitoring. The present study revealed the ability of both EOs to control *M. euphorbiae* populations under greenhouse conditions and hinted at the importance of assessing EO efficiency in realistic agronomic conditions (e.g., under the fluctuating environmental conditions usually occurring in greenhouses).

**Keywords:** insecticidal/aphicidal activity; botanical pesticides; pest management; aphids; fumigants; phytotoxicity

## 1. Introduction

Aphids are found worldwide and represent one of the most common crop pests in temperate areas. Approximately 450 species affect economically important crops such as corn, wheat, potatoes and tomatoes [1,2]. Since post World War II, chemical control is by far the method most used to minimize aphid threats to crops [3]. However, damaging side effects of chemical pesticide usage are becoming increasingly apparent. Notable risks involved in pesticide usage include a risk of toxicity to the natural enemies of pests and to farmers and consumers, in addition to a risk of resistance development [4–7]. Its negative impact on both human health and the environment have recently led to changes in European pesticide regulations, resulting in the ban of many chemical insecticides, including neonicotinoids [8].

The transition toward a safer, more ecological alternative has led to an interest in a more holistic pest-control strategy known as Integrated Pest Management (IPM). Biopesticides are part of IPM. They are a promising, environmental-friendly alternative means of

pest control based on the use of living organisms or natural materials derived from animals, plants, bacteria, fungi or minerals [9]. Currently, biopesticides account for approximately 5% of the global crop protection market. This unimpressive percentage is due to the lack of time and money invested in developing and commercializing new "green" crop protection products [10]. However, with a growing demand both from consumers and producers, biopesticides are becoming a bright alternative to synthetic insecticides [11].

Among the potential biopesticides, essential oils (EOs) represent one of the most promising options. EOs are hydrophobic liquids essentially composed of volatile compounds belonging mainly to the phenylpropanoid (e.g., trans-anethol) and terpenoid (e.g., monoterpene and sesquiterpene) families [12]. To be classified as EOs, they must be obtained from plant raw material by steam distillation, by mechanical processes (e.g., epicarp of citrus fruits) or by dry distillation, after separation of the aqueous phase [13]. The compounds they contain include plant defense chemicals which provide extensive natural protection against plant pests e.g. EOs can have a neurotoxic mode of action [14,15] that can be lethal to insects. They can also act to repel and deter pests and to prevent feeding, reproduction, oviposition and development [16,17]. The environmental risks of EOs are limited by their rapid conversion and degradation by oxygen, moisture, light and heat and have proven non-toxic to mammals in studies with culinary, cosmetic and medicinal applications, opening the door to their use as eco-friendly biopesticides [18]. EOs that contain certain compounds, such as monoterpenes, have been shown to provide better insecticidal efficacy [17]. Another major factor that determines EO toxicity is the proportion of these components in an EO. This proportion differs between aromatic plant species [18]. Numerous studies have shown their effectiveness against aphids. Extracts from the Lamiaceae, Asteraceae or Apiaceae [19] families have unquestionably proven to be efficient against aphids at the laboratory scale, when applied by both contact and fumigation. *Foeniculum vulgare* (fennel) and *Pimpinella anisum* (green anise), which both belong to the Apiaceae family, have excellent toxic activity against a broad spectrum of aphids including *Lipaphis pseudobrassicae*, *Myzus persicae*, *Acyrthosiphon pisum*, *Brevicoryne brassicae*, *Aphis gossypii* or *Aphis fabae* [20].

The efficiency of EOs as aphicidal products is undeniable under laboratory conditions [20]. However, all too often, observations assessed in laboratory experiments are not replicable on a larger scale. Promising products identified in the lab often prove to be unusable on crops in real-world settings. Few trials have previously been conducted in a greenhouse or in the field to test potential biopesticides. Nonetheless, from what is known, using EOs as biopesticides against aphids under field conditions should be feasible. Soliman, (2007) [21] described the great potential of two pulverized EOs (*Artemisia herba-alba* and *Artemisia monosperma*) against the cotton aphid, *Aphis gossypii*, finding a considerable reduction of population size in treated greenhouses even 15 days after the first contact application. This assessment was also confirmed by data from Pacific AgResearch reported in Isman et al. (2011) as unpublished data [22]. In a field study, they showed important control of *Myzus persicae* by an essential oil-based insecticide sprayed on crops, which proved comparable in efficiency to synthetic insecticides.

More large-scale experiments are necessary to ascertain the insecticidal capacity of EOs at an agricultural-relevant scale. The verification of EO control at the crop scale is a key step toward the adoption of EOs as biopesticides. In this study, we assessed the contactless entomotoxic effect of green anise and bitter fennel EOs against the potato aphid, *Macrosiphum euphorbiae*. The experiment was conducted in three settings of increasing size (i.e., leaflet, whole plant and greenhouse) to determine the consistency of EO control from the lab to the agronomic scale. First, leaflet bioassays were used to plot dose-response curves for fennel and green anise EOs and determine the mean lethal concentration ($LC_{50}$). Second, whole-plant experiments tested the $LC_{50}$ values previously determined in leaflet experiments. Third, greenhouse experiments focused on the aphid population dynamics after EO fumigation. The potential phytotoxic effect of EOs was monitored for all experiments.

## 2. Material and Methods

### 2.1. Biological Material

#### 2.1.1. Plants

All experiments were performed on *Solanum lycopersicum* L. tomato plants of the Nano variety. Plants used in laboratory bioassays were grown in a climatic chamber (24 ± 2 °C, 40 ± 10% RH, 16:8 L.D.) for 4 to 5 weeks before being moved to the greenhouse (25 ± 6 °C, 60 ± 15% RH, seasonal photoperiod). Leaflets used in laboratory experiments were collected on plants aged minimum 8 weeks. Plants used in greenhouse experiments were moved from the climatic chamber 1 week after seedling to acclimate them to greenhouse conditions.

#### 2.1.2. Insects

Individuals that naturally infest tomato plants were used to initiate the colony of *Macrosiphum euphorbiae* (Thomas) (Hemiptera: Aphididae) at the National Institute for Agronomic Research & Environment (INRAE) Sophia Antipolis experimental station (Southeastern France). Aphids used in this study were reared on Nano variety tomato plants in the INRAE greenhouses, Sophia Antipolis, France (24 ± 5 °C, 60 ± 20% RH, seasonal photoperiod). All experiments were performed on individuals in the second or third nymphal stages.

### 2.2. Chemical Materials and Analysis

The essential oils selected were those found to provide the greatest target entomotoxic effect against *M. euphorbiae* without inducing important phytotoxic damage on treated plants in a pre-experiment (Figure S1). Bitter fennel (*Foeniculum vulgare* Mill.) and green anise (*Pimpinella anisum* L.) EOs were purchased from Esperis s.p.a (Milan, Italy).

#### 2.2.1. Gas Chromatography-Mass Spectrometry (GC-MS)

Prior to analysis, both EOs were diluted with pentane to 80 mg/mL. GC/MS analyses were carried out using an Agilent 6890N/5973N system equipped with an HP-1 (0.2 mm × 50 m × 0.33 µm) column. The carrier gas was helium in constant flow mode at 1 mL/min. The split ratio was 1/100 with an injection volume of 1 µL. The injector and transfer line were set at temperatures of 250 °C and 270 °C, respectively. Oven temperature was set to 40 °C and subsequent to injection temperature, was increased by 2 °C per minute until it reached 200 °C (80 min). The temperature was then increased another 20 °C per minute until it reached the final temperature of 270 °C which was maintained for 5 min, providing a total analysis time of 88.5 minutes. Acquisition was performed in scan mode (35–400 a.m.u./sec) and mass spectra were generated at 70 eV.

#### 2.2.2. Compound Identification

Retention indices were determined with reference to a series of linear alkane standards (C6–C28). Identification of the compounds was based on computer matching against commercial libraries (i.e., Wiley, NIST), laboratory mass spectra libraries (i.e., Flora97, Lca98) containing pure substances and MS literature data [23,24], combined with comparisons of GC linear retention indices.

### 2.3. Experimental Set-Up

Three treatments were designed and used at all experimental scales; one control treatment and two EO treatments. Green anise and fennel EOs were used in their pure form and were applied to ashless Whatman™ filter paper for non-contact fumigation treatments. As control, water was applied to filter paper.

#### 2.3.1. Leaflet Bioassays

Dose-response curves were plotted and the mean lethal concentration (LC$_{50}$) for each of the two EOs was determined in leaflet bioassays conducted under laboratory conditions

(22 ± 3 °C, 45 ± 15% RH, seasonal photoperiod). A single leaflet was suspended inside an inverted clear plastic cup (500 mL, height: 10 cm). Its stem was inserted through a hole in the middle of the cup's base and secured in a 1.5 mL Eppendorf tube filled with water (Figure 1A). Ten second-to-third-instar aphid individuals were then added to the experimental system. Twelve replicates were carried out per treatment (i.e., 120 aphids in total). The system was then closed by means of a mesh square held by a rubber band. This system was placed above an open Petri dish (diameter: 9 cm) containing a Whatman™ filter paper to which was applied one of the tested EOs or water for the control treatment.

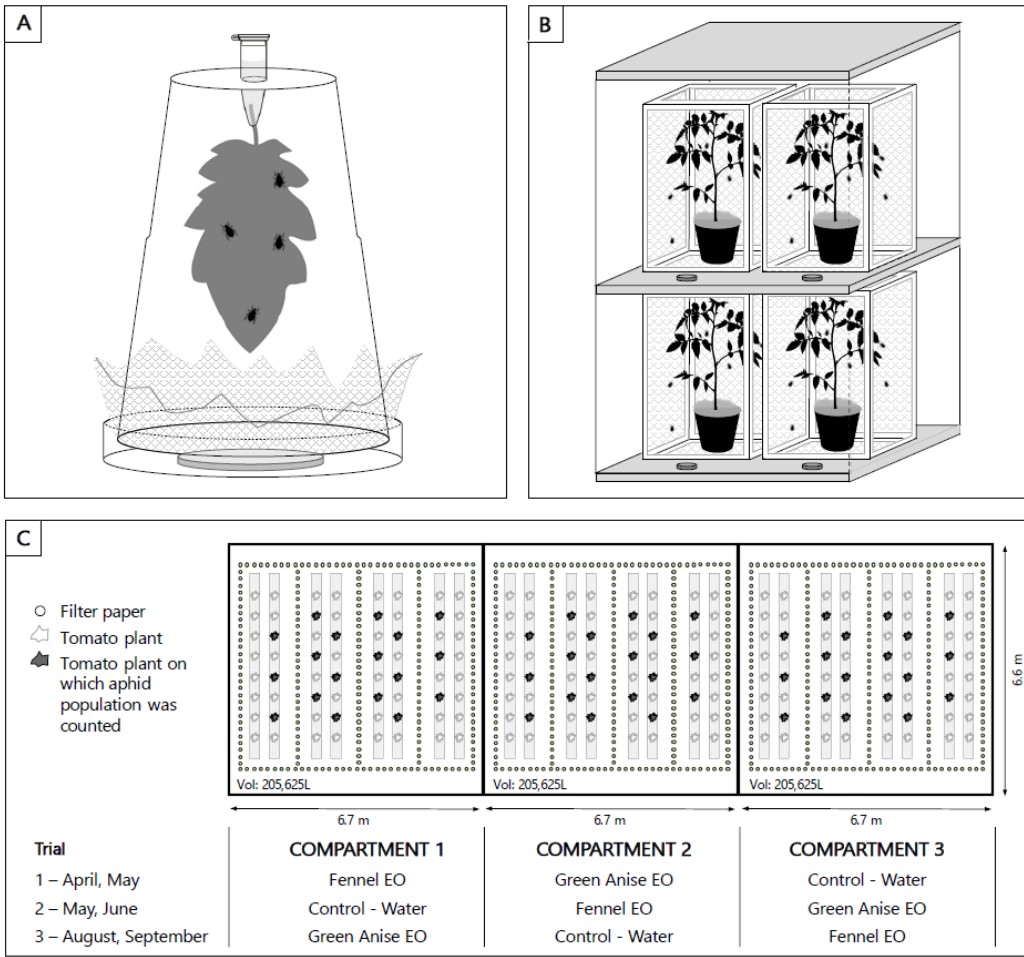

**Figure 1.** Experimental designs; (**A**): Leaflet bioassays, (**B**): Whole plant bioassays. The figure represents the distribution of four cages inside the climatic chamber. The small cylinders show the 8 Petri dishes (4 in front and 4 behind the cages) containing the fumigated substance used for the experiment, (**C**): Greenhouse experiment. Diagram of compartments with the plant arrangement–counted plants are highlighted in dark grey. Circles represent filter papers placed between plant rows. The distribution of treatments between compartments and time period trials (R) are shown under the diagram.

A range of concentrations (i.e., 2, 5, 10, 15, 20 and 30 μl L$^{air}$) of each EO was tested by fumigation, beginning with the range used in the preliminary screenings and extending above and below so as to obtain complete dose-response curves. After EO application, the system was left in place for 24 h, after which time the number of living and dead individuals in the system was counted and the mortality rate was calculated, according to the formula below.

$$Mortality\ rate = \frac{Number\ of\ dead\ individuals\ after\ 24\ h\ of\ EO/control\ treatment}{Total\ number\ of\ individuals}$$

Calculated mortality rates and the corresponding concentrations of EO were used to plot dose-response curves and calculate the $LC_{50}$ for each EO.

Subsequent to this calculation, $LC_{50}$ toxicity for each EO was tested again on *M. euphorbiae* (in the same conditions as described above) to confirm its validity. The determined concentration of each EO (green anise and fennel) was applied to the inverted cup experimental system into which 10 *M. euphorbiae* individuals were introduced. Twelve replicates were carried out per treatment (i.e., 120 aphids in total). Mortality rates were determined 24 h later and compared with the expected $LC_{50}$ values.

The effect of EO fumigation on the plant was also recorded systematically. A visual six-level index, defined by the team, was used: 0 = healthy plant/leaflet, 1 = less than 5% of the plant/leaflet necrotized, 2 = between 6 and 25% of the plant/leaflet necrotized, 3 = between 26 and 50% of the plant/leaflet necrotized, 4 = between 51 and 75% of the plant/leaflet necrotized and 5 = between 76 and 100% of the plant/leaflet necrotized. Phytotoxicity was assessed 24 h after substance application.

### 2.3.2. Plant Experiments

Plant experiments used to test the $LC_{50}$ values determined by the leaflet bioassays were conducted in climatic chambers under controlled conditions (volume: 543 $L^{air}$, 24 $\pm$ 1 °C, 70 $\pm$ 10% RH, 14:10 L.D.) on 4-to-5-week-old tomato plants. One tomato plant was placed in each nylon cage (28 cm $\times$ 28 cm $\times$ 42 cm) and four cages were placed in each climatic chamber, two per shelf (Figure 1B). Thirty second-to-third-instar individuals were placed in each cage. Twelve replicates were carried out per treatment (i.e., 360 aphids in total). EOs were applied to Whatman$^{TM}$ filter paper placed in glass Petri dishes outside the cages, one behind and one in front of each cage. The quantity of EO applied corresponded to the previously determined $LC_{50}$, adjusted to the volume of the climatic chamber ($LC_{50} \times 543$ $L^{air}$). This volume was equally divided between all filter papers inside the climatic chamber. For the control treatment, water was applied to filter papers. As the insects were inside the cages, no direct contact was possible between the insects and the EOs. Only EO volatile molecules passing into the cages through fumigation came in contact with the aphids.

Mortality was determined 24 h after EO treatment application and phytotoxicity was recorded as described previously.

### 2.3.3. Greenhouse Experiments

To determine whether the results of the laboratory experiments could be extrapolated to a more realistic agronomic scale, greenhouse experiments were performed to test the entomotoxic and phytotoxic effects of the two EOs. The greenhouse experiments were conducted in greenhouse compartments at the INRAE, Sophia Antipolis, France (compartment size: 44 $m^2$, volume: 205,625 $L^{air}$, 24 $\pm$ 2 °C, 45 $\pm$ 15% RH, seasonal photoperiod). Each greenhouse compartment was submitted to one treatment: control/water, green anise EO or fennel EO. This experimental set-up was repeated three times in order to avoid a compartment effect as treatments were moving from one compartment to another in the different repetitions (Figure 1C). The three trials were carried out over the spring and summer seasons of 2019: Trial 1 took place from 15 April to 7 May, trial 2 from 20 May to 11 June and trial 3 from 12 August to 3 September.

To recreate realistic crop conditions and reproduce standard plant density, 64 six-week-old tomato plants were placed in each compartment (Figure 1C). Ten second or third-instar *M. euphorbiae* nymphs were placed on each plant, one day before the EO treatment. As a result of the high phytotoxicity observed on tomato plants in climatic chamber experiments with fennel EO, a reduced quantity of fennel EO was applied, corresponding to an $LC_{18}$ equaling the green anise EO quantity (1.4 L). As no phytotoxicity damage was recorded for green anise EO in smaller-scale experiments, its $LC_{50}$ (i.e., 1.4 L) was applied in the greenhouse. EOs were applied to filter paper evenly spaced on the greenhouse floor between the rows of plants during a 24-h exposure period during which all openings were

locked shut (Figure 1C). After 24 h, the filter papers were removed from the compartments which were ventilated for 24 h. Aphids on 18 of the 64 plants per compartment were counted 2, 7, 13, 17 and 21 days after EO application.

Phytotoxicity was determined via visual assessment of the overall plant. This effect was then recorded using the same visual six-level index as in the leaflet and plant experiments. Phytotoxicity was monitored over the 21-day period starting 2 days after EO application and on counting days.

### 2.3.4. Statistical Analysis

At the leaflet level, the $LC_{50}$ was estimated by plotting dose-response curves using a probit analysis (function "dose.p", package "MASS").

In leaflet and whole-plant experiments, Generalized Linear Models (family = binomial) were used to analyze differences in mortality between treatment groups (control, green anise EO application, fennel EO application). Significant treatment effects were followed up by Tukey's HSD post hoc test (function "lsmeans", package "lsmeans").

In greenhouse experiments, population dynamics were analyzed by fitting a Generalized Estimating Equation (or GEE, package "geepack") with a Poisson distributed response. A GEE was used to take into account the effect of time and the interdependence of data. P-values were adjusted using the Bonferroni adjustment method.

A non-parametric Kruskal-Wallis test was used to assess EO phytotoxicity on tomato plants between treatments. Non-parametric multiple comparisons were performed using a Dunn test with a Bonferroni adjustment. In greenhouse experiments, phytotoxicity dynamics were also analyzed using a GEE ("geepack" package) with a binomial distributed response. P-values were adjusted in the same way as for population dynamics in multiple comparison tests.

All statistical analyses were completed using R statistical software (version 3.5.3).

## 3. Results

### 3.1. Chemical Analysis of the Essential Oil Composition

Green anise EO shows a nearly monomolecular composition and contains 87% trans-anethole, a propenylbenzene compound (Table 1). The remainder of its composition consists of a variety of other volatile organic compounds present in insubstantial quantities. Fennel EO shows a more diversified profile, composed of only 41% trans-anethole, complemented by 32% limonene, 10% fenchone and various minor components (Table 1).

**Table 1.** Composition of the green anise and fennel EOs by GC-MS. Retention indices (RI) are determined on an HP-1 column using the homologous series of n-alkanes (C6–C24). RIs from the literature were obtained from the NIST database.

| Compounds | Measured RI | Literature RI | Green Anise (%) | Fennel (%) |
|---|---|---|---|---|
| α-pinene | 932 | 932 | 0.58 | 2.38 |
| camphene | 953 | 954 | | 0.29 |
| sabinene | 969 | 967 | 0.04 | 0.12 |
| β-pinene | 974 | 977 | 0.14 | 0.9 |
| myrcene | 988 | 985 | 0.08 | 1.07 |
| α-phellandrene | 1002 | 1003 | 0.29 | 2.73 |
| δ-3-carene | 1008 | 1011 | 0.16 | 0.28 |
| α-terpinene | 1018 | 1017 | | 0.37 |
| p-cymene | 1020 | 1025 | 0.04 | |
| limonene | 1024 | 1028 | 1.14 | **31.53** |
| β-phellandrene | 1025 | 1030 | 0.32 | |
| cis-β-ocimene | 1040 | 1037 | | 0.51 |
| γ-terpinene | 1054 | 1058 | 0.16 | 0.18 |
| Unidentified | 1082 | | 0.1 | 1.74 |
| fenchone | 1083 | 1071 | | **10.1** |
| terpinolene | 1086 | 1087 | 0.05 | 0.2 |

**Table 1.** *Cont.*

| Compounds | Measured RI | Literature RI | Green Anise (%) | Fennel (%) |
|---|---|---|---|---|
| linalool | 1095 | 1098 | 1.86 | 0.61 |
| camphor | 1141 | 1145 | 0.03 | 0.15 |
| terpinene-4-ol | 1174 | 1174 | 0.14 | 0.13 |
| α-phellandrene epoxide | 1179 | 1187 | | 0.18 |
| α-terpineol | 1186 | 1190 | 0.11 | |
| estragol | 1195 | 1197 | 3.91 | 2.45 |
| anisaldehyde | 1247 | 1252 | 0.52 | 0.52 |
| trans-anethole | 1249 | 1264 | **87.3** | **41.49** |
| cis-anethole | 1282 | 1284 | 0.23 | 0.21 |
| α–copaene | 1374 | 1375 | 0.05 | |
| eugenol | 1351 | 1358 | | 1.05 |
| β-caryophyllene | 1407 | 1419 | 0.3 | 0.18 |
| α-bergamotene | 1411 | 1430 | 0.37 | 0.15 |
| β-farnesene | 1454 | 1459 | 0.02 | |
| γ-muurolene | 1478 | 1497 | 0.01 | |
| valencene | 1496 | 1496 | 0.02 | |
| α–farnesene | 1505 | 1508 | 0.05 | |
| α–bisabolene | 1506 | 1505 | 0.06 | |
| δ-cadinene associate | 1522 | 1522 | 0.05 | |
| nerolidol | 1561 | 1565 | 0.06 | |
| T-muurolol | 1644 | 1641 | 0.06 | |
| foeniculin | 1677 | 1684 | 1.07 | 0.46 |
| linalyle benzoate | 2157 | na | 0.66 | |

### 3.2. Fumigation Treatments at Laboratory Scale–Dose-Response Curves and Determination of the $LC_{50}$

Fitted dose-response curves were found to be different between green anise EO and fennel EO. $LC_{50}$ values equaled 6.6 µl L$^{-1}$air and 12.2 µl L$^{-1}$air for green anise and fennel EOs, respectively (Figure 2).

Subsequent bioassays confirmed the accuracy of both $LC_{50}$ values (mortality rate: 53%-Green anise EO; 55%-Fennel EO) (Figure 3-Leaflet). Aphid mortality varied significantly depending on the treatment ($X^2_2 = 53.47$, $p < 0.001$). Both EOs induced a higher mortality compared to the control treatment (control–green anise: $p < 0.001$, control–fennel: $p < 0.001$) but mortality did not differ between the two EO treatments ($p = 0.575$). Phytotoxicity was significantly different between treatments ($X^2_2 = 25.268$, $p < 0.001$).

No phytotoxicity was recorded in the control treatment (Figure 4A-Leaflet). Green anise EO impact on the plant was not significantly different from the control condition ($p = 0.536$). Harmful effects were assessed on the tomato leaflet throughout the fennel EO treatment compared to the control treatment ($p < 0.001$).

### 3.3. Fumigation Treatments at the Whole-Plant Scale–Verification of the $LC_{50}$ at the Whole-Plant Scale

Whole-plant experiments on *M. euphorbiae* allowed a plant-scale verification of the results obtained at the leaflet level. Both EOs induced significantly greater mortality than the control treatment ($X^2_2 = 104.16$, $p < 0.001$) (Figure 3–Whole Plant). The mortality rate shifted from 53% to 46% between scales for the green anise EO and from 55% to 68% for fennel EO, showing a significant difference between EO treatments ($p < 0.001$). (Figure 3). Phytotoxicity was significantly different between treatments ($X^2_8 = 145.91$, $p < 0.001$). No phytotoxicity was recorded from the green anise EO treatment ($p > 0.001$) (Figure 4A–Whole Plant) contrary to the fennel treatment ($p < 0.001$). Damage assessed at the whole-plant level was significantly higher than in the leaflet experiment ($p < 0.001$).

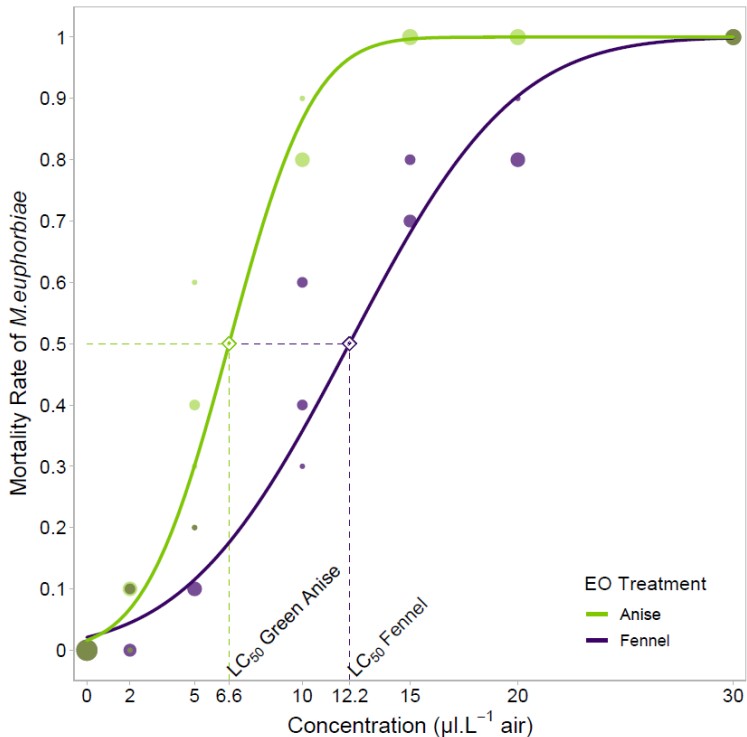

**Figure 2.** Dose-response curves for *M. euphorbiae* in response to increasing concentrations of green anise and fennel EOs (from 0 to 30 µl L$^{-1}$ air) applied by fumigation, and the subsequently calculated LC$_{50}$ values. Control (no EOs) is shown by point (0;0).

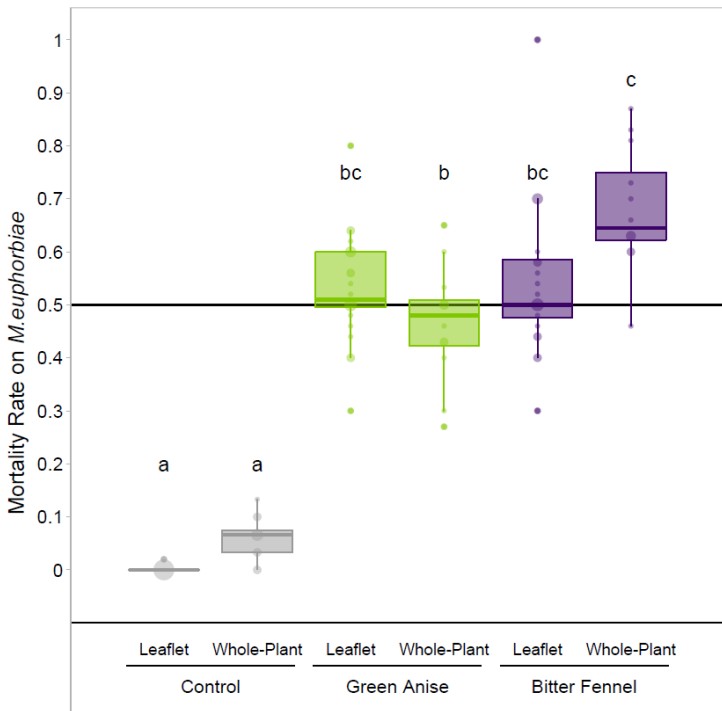

**Figure 3.** Mortality rate of *M. euphorbiae* in leaflet and whole-plant trials when exposed to the calculated LC$_{50}$ values of green anise and fennel EOs. Different letters indicate significant differences between groups ($X^2_2 = 104.16$, $p < 0.001$).

### 3.4. Fumigation Treatments at the Greenhouse Scale–Population Dynamics

Greenhouse experiments evaluating the effect of EO fumigation treatments on *M. euphorbiae* population growth, over a 21-day period, resulted in a significantly smaller population size with application of either green anise or fennel EOs when compared with the control treatment (Figure 5; $X^2_2 = 140$, $p < 0.001$).

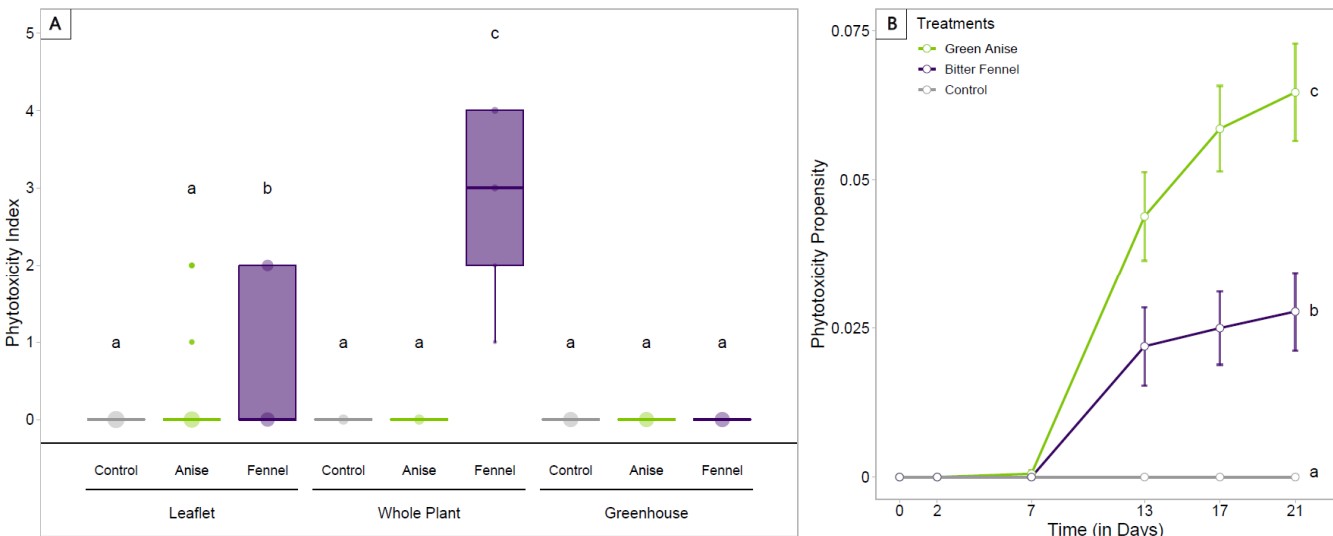

**Figure 4.** Recorded phytotoxicity on tomato plants in leaflet (**A**), whole-plant (**A**) and greenhouse experiments (**B**). (**A**): Phytotoxicity was assessed after 24 h of exposure to EOs. Different letters indicate significant differences between groups ($X^2_8 = 145.91$, $p < 0.001$). (**B**): Phytotoxicity propensity dynamics over a 21-day period after application of 1.4 L of either green anise (LC$_{50}$) or fennel EO (LC$_{18}$) in greenhouse compartments, compared to a non-treated control compartment. Error bars indicate SE. Different letters indicate significant differences between groups ($X^2_2 = 135$, $p < 0.001$).

No difference in population size between the two EOs was observed, with green anise and fennel EOs displaying an equal capacity to limit population growth when applied in equal quantities ($p = 0.2777$) (Figure 5). This observation contrasts with the expectation of a greater population control by green anise EO, given we applied its LC$_{50}$, compared with the LC$_{18}$ only for fennel EO.

Seasonal differences resulted in variations in the abiotic conditions between trials as conditions could only be minimally tempered by the cooling and shading mechanisms of the greenhouses used. The average temperature increased across trials, with temperatures averaging $24 \pm 3$ °C, $26 \pm 3$ °C, and $29 \pm 2$ °C, in trials 1, 2 and 3, respectively. Humidity also varied between the trials, with an overall increase from trial 1 to 3: $58.2 \pm 17\%$ RH (trial 1), $61.4 \pm 13\%$ RH (trial 2), and $75 \pm 10\%$ RH (trial 3). Despite these differences in abiotic conditions, the difference between EO treatments and the control treatment was significant and EO effects were similar (less efficiency for fennel EO in summer conditions; Trial 1–Trial 3: $p < 0.001$, Trial 2–Trial 3: $p < 0.001$).

Phytotoxicity significantly varied between treatments ($X^2_2 = 135$, $p < 0.001$) (Figure 4A-Greenhouse). The EO treatment started producing burns on the plants' oldest leaves 13 days after fumigation (i.e., leaves that were already fully developed during the 24-h EO fumigation). The green anise treatment (LC$_{50}$) produced twice the phytotoxicity of the fennel EO treatment (LC$_{18}$) ($p < 0.001$). The control treatment did not show any phytotoxicity during the entire experiment (control–green anise: $p < 0.001$, control–fennel: $p < 0.001$) (Figure 4B).

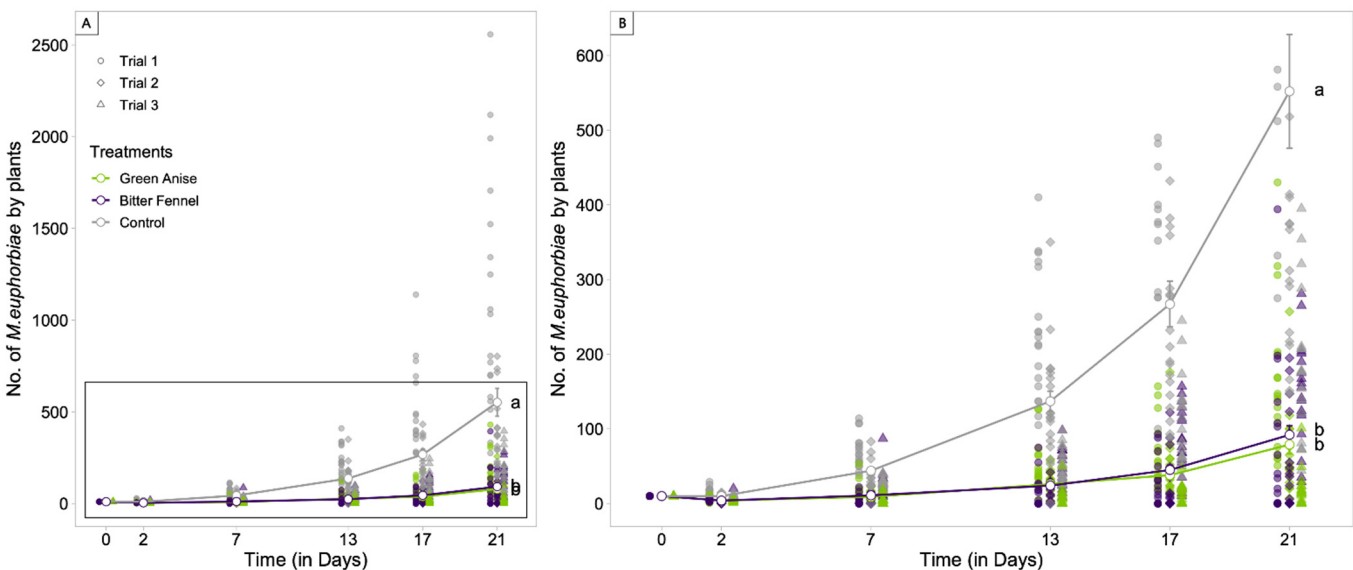

**Figure 5.** *M. euphorbiae* population dynamics over 21 days; number of *M. euphorbiae* individuals per tomato plant over a 21-day period after application of 1.4 L of either green anise (LC50) or fennel EO (LC18) in a greenhouse compartment, compared to a non-treated control compartment. Error bars indicate SE. Different shapes indicate different trials and colors illustrate the different treatments. Different letters indicate significant differences between groups ($X^2_2 = 140$, $p < 0.001$). (**B**) is a zoom of graph (**A**).

## 4. Discussion

Our study confirms the potential of EOs as a biopesticide/insecticide applied by fumigation to control aphids in an agronomic greenhouse setting. It is, however, clear that our two EOs do not have the same efficiency and variability in population control can be observed between trials. Although a promising alternative to synthetic pesticides, EOs can also be harmful to crops and this calls for caution and more studies.

### 4.1. Across-Scale Assessment of the Efficiency of EO Treatments

It has been stated multiple times as a concluding point that even though EO efficiency in the lab is undeniable, confirmation in field settings is lacking [2,12,25]. The results of this study demonstrate the ability of both tested EOs (green anise and fennel) to control *M. euphorbiae* populations at the leaflet, whole-plant, and greenhouse scale, and establish the agriculturally relevant biocontrol potential of these two natural substances. Leaflet and whole-plant experiments allowed us to define and test the LC$_{50}$ values of green anise and fennel EOs, which proved equal to merely 6.6 and 12.2 µl L$^{-1}$air, respectively. When 6.6 µl L$^{-1}$air (LC$_{50}$ for green anise and LC$_{18}$ for fennel equaling 1.4L for a greenhouse compartment) of both EOs were applied at the greenhouse scale, this resulted in an average observed reduction in *M. euphorbiae* population size of more than 80% for both EOs, corresponding to a population size post-EO treatment equal to 20% of that observed in the control. This impressive control, which was maintained for 21 days after a 24-h application, shows the consistent efficiency of the two EOs across scales, making them promising biopesticide candidates.

Nonetheless, EOs caused damage to the tomato leaves as already reported for other Apiaceae EOs on crops [19]. In the leaflet and whole plant experiment, the green anise EO generated lower phytotoxicity on the plants than fennel EO fumigation. Whole-plant experiments showed an increased phytotoxic effect of fennel EO. Interestingly, by reducing the fennel EO concentration in the greenhouse experiment (i.e., applying LC$_{18}$), the objective of avoiding a high level of phytotoxicity was achieved without reducing fennel EO efficiency against *M. euphorbiae* compared to the green anise EO treatment. However, both EOs produced burns on leaves from day 13 after fumigation. The delayed effect on plant

health had already been described in Digilio et al. (2008) [26] but in that study the plants exposed to EO vapors developed burns three days after EO fumigation.

The present study stresses and reinforces the importance of long-term monitoring of crops after EO application, particularly the assessment of the phytotoxic impact of EOs on crop yield. One of the highlights of our study was that even when we applied a reduced fennel EO concentration in the greenhouse experiment (from $LC_{50}$ to $LC_{18}$), no loss in entomotoxic efficiency was observed when phytotoxicity was reduced (compared to leaflet and whole-plant experiments).

### 4.2. Relative Efficiency and Variability between Green Anise and Fennel EO Treatments

EOs are complex mixtures characterized by the presence of volatile compounds belonging to either the terpene or the phenylpropanoid chemical classes [12,27]. Generally, they are dominated by a limited number of major compounds presumably often considered as those involved in the biopesticide effect [20]. Green anise EO and Fennel EO have quite different chemical compositions. Green anise EO is composed of 87% of one main constituent: trans-anethole whereas fennel EO has greater compositional variation, containing 41% trans-anethole, 32% limonene and 10% fenchone (Table 1). It is interesting to note that fennel EO contains a percentage of trans-anethole equal to roughly one-half the percentage present in green anise EO. The literature has already shown the implication of anethole, limonene and fenchone in EO toxic activity against aphids [19,20]. One hypothesis to explain the EO aphidicidal effect is that compounds inhibit the activity of acetylcholinesterase (AChE) [14,19]. AChE is a key enzyme involved in neuro-neuronal and neuromuscular junctions in insects. It has a neurotoxic effect on aphids, leading to paralysis and death. However, this would require high concentrations. A broader explanation would be that because volatile compounds are lipophilic, they could disrupt aphid cell membranes, as observed on pathogens and other insects [27–29]. Oxidized monoterpenes (e.g., trans-anethole) presumably confer toxicity because of their lipophilic properties [30]. As stated before, both green anise and fennel EOs contain oxidized monoterpenes and have proven to be efficient against *M. euphorbiae* in this study but also against many other aphid species [20]. The relative efficiency of the two EOs, however, was shown to vary across the scales tested. At the leaflet scale, green anise EO proved nearly twice as efficient as fennel EO ($LC_{50}$ of 6.6 µl $L^{-1}$air for green anise EO versus $LC_{50}$ of 12.2 µl $L^{-1}$air for fennel EO). However, this pattern was progressively reversed with an increase in scale.

### 4.3. Potential Causes of EOs Efficiency Variability

Differences in population control between green anise and fennel EOs, when applied in the same quantity at the greenhouse scale, proved non-significant, despite the greater efficiency of green anise EO observed in laboratory experiments. This increase in fennel EO efficiency associated with an increase in experimental scale suggests a potential biostimulatory effect of this EO on the plants tested. In addition to its direct entomotoxic effect on *M. euphorbiae*, fennel EO may stimulate defense responses when applied to whole plants with an intact metabolism, thus resulting in an increase in plant defense responses to *M. euphorbiae* infestation. The literature has shown that the emission of Volatile Organic Compounds (VOCs), especially monoterpenes, can trigger the priming of plant defenses by inducing the transcription of defense genes via the jasmonic acid pathway [31]. VOCs can also induce the production and emission of monoterpenes from plants [32]. In a more specific way, it has been proven that thyme EO can enhance tomato plant defenses against *Botrytis cinerea* and other fungi [33].

This effect may be further enhanced by the microclimate created in the greenhouse by the density of plants used. The great number of plants in close proximity may have led to greater plant transpiration, resulting in modifications of the temperature and humidity between plants, affecting the vaporization of EOs and their resulting effects on the plants and their pests [31]. More important differences in temperature and humidity between greenhouse trials resulted from seasonal changes may also help to explain the differences

in EO control efficiency observed. Experiments performed at the greenhouse scale were subject to seasonal variations in abiotic conditions as the greenhouse cooling system could only temper external conditions to a minimal extent. The first greenhouse trial was performed from mid-April to early May, whereas the third was performed from mid-August to early September, resulting in a 5 °C increase in the average temperature and a roughly 20% increase in relative humidity. These changes in the abiotic condition coincide with a significant decrease in fennel EO control efficiency. One possible explanation is that even though all compounds have an increased volatility at high temperature, high humidity can either increase or decrease compound emission. Vallat et al. (2005) [34] showed that an increase in relative humidity had a negative impact on the limonene emissions from apple trees. They also found that ketone compounds follow the same trend. Qifan et al. (2017) [35] also observed a reduction in emissions of ketone compounds from plywood due to an RH increase. Fennel EO contains limonene and 10% of fenchone which has a ketone function. Finally, differences in efficiency between the three trials can also be explained by the change in temperature. Pavela & Sedlák (2018) [36] showed in a contact application study on *Culex quinquefasciatus* that an increase in the post-application temperature can result in a decrease in efficiency.

Whatever the explanation, it is clear that if fennel EO is to be used as an effective biopesticide, greenhouse conditions must be maintained around 25 °C and 60% RH. Further experiments need to be performed to determine the abiotic conditions at which green anise and fennel EOs provide the most efficient pest control.

## 5. Conclusions

At the greenhouse scale, EOs caused an 80% decrease in the *M. euphorbiae* population size, compared to the control treatment maintained over a period of 21 days, highlighting a potential use of EOs as biopesticides at agronomic scale (greenhouse). Several factors explaining EO efficiency variability were also suggested (e.g., EO composition, application scale and abiotic conditions).

Even so, more studies will need to be carried out before EOs can rival synthetic pesticides. Firstly, the effect of EOs on crop yield needs to be examined in order to assess the balance between cost (phytotoxicity) and benefits (aphid control). Secondly, the mode of action of EO fumigation on both target (pest) and non-target (crops) is still uncertain. Thirdly, the responsibility of each EO compound in the aphicidal effect needs to be assessed in order to optimize future biopesticide products based on EOs as well as the potential for developing resistance.

As of today, only a few EO-based biopesticides are commercialized and only one has been approved for use in the EU (i.e., Prev-Am) [37,38]. As reviewed in Isman (2020), these biopesticides can either be composed of a mix of ingredients or of a single one (i.e., Prev-Am only has orange EO as an active ingredient) and none are designed for a fumigated application [37]. In order to market EOs as a biopesticide applied by fumigation, formulation will be essential. Improved biopesticide formulation would allow for greater stability, safety, ease of application and persistence of effect [39]. Recent studies have demonstrated the benefits of formulation for EO-based biopesticides, including increased dispersion, improved stability and a more persistent release, most notably for the release of active terpenes [40–42].

In a study by Soares et al. 2019 [38], the use of an association of Prev-Am, an EO-based biopesticide, with biological control has proven more efficient against *Tuta absoluta* than the use of a synthetic pesticide. The benefit of the combined association of both EO fumigation and biological control is still an open question. In preliminary results, we observed greater sensitivity of an aphid parasitoid to EO fumigation compared to aphids whereas predators were less sensitive (Dunan, Malanga et al. non-published data).

Formulation of green anise or fennel EOs would allow for their use by fumigation and this method could potentially be paired with other IPM methods to enhance control over aphid populations, but only if the phytotoxicity side-effect is controlled.

**Supplementary Materials:** The following are available online at https://www.mdpi.com/article/10 .3390/agriculture11090867/s1, Figure S1: Mortality rate of M. euphorbiae in response to a 16μl/L air concentration of different EOs compared to the control treatment. Different letters indicate significant differences between groups (F = 121.8, df = 5, *n* = 6, *p*-value < 0.001).

**Author Contributions:** Conceptualization, A.-V.L., N.D. and T.M. (Thomas Michel); funding acquisition, A.-V.L. and N.D.; methodology, A.-V.L., N.D. and T.M. (Thomas Michel); conduction of experiments L.D., T.M. (Tara Malanga) and P.B.; conduction of suppl. experiments S.B. and P.B.; formal analysis, L.D. and L.S.M., T.M. (Tara Malanga), A.-V.L. and T.M. (Thomas Michel); writing—original draft preparation, L.D. and T.M. (Tara Malanga); writing—review and editing, L.D., T.M. (Thomas Michel) and A.-V.L. All authors have read and agreed to the published version of the manuscript.

**Funding:** The experiments presented received financial support from the project "Sustainable Tomato Production" (STomP–ARIMNet2 618127-EU) and from the EUCLID project (H2020 no. 633999).

**Data Availability Statement:** The raw data of our paper can get access: https://unice-my.sharepoint.com /:x:/g/personal/anne-violette_lavoir_unice_fr/EcIbaZYvXFRGubC_EbI1HowB39PwF6kgN6OJnNAxah Slvg?e=LKOibt (accessed on 9 September 2021).

**Acknowledgments:** We are very grateful to Lionel Salvy, Richard Brun, Edwige Amiens, Christiane Métay-Merrien, for lending a helping hand and sharing their expertise during the experiment presented.

**Conflicts of Interest:** The authors declare no conflict of interest.

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
