# Peer review of "Biopesticide Evaluation from Lab to Greenhouse Scale of Essential Oils Used against Macrosiphum euphorbiae"

_agriculture, doi:10.3390/agriculture11090867_

Round 1
Reviewer 1 Report
Dear Authors,
I read your interesting paper with pleasure and I have to make my own compliments for this interesting topic, being an alternative way to control insect pests such as aphids, instead of traditional insecticides. The literature is updated and the experimental plan and results well described. Anyway, I found some mistakes and contradictions in the text. Please try to follow my recommendations.
Lines 28-30: erase this sentence; is not appropriate for the abstract
Line 101: Material instead of materiel
Lines 117-118: rewrite the sentence that is not so clear
Lines 122-123: what kind of formulation?
Line 144 onwards: are you sure that you could use the term fumigation in this experiment? I think that in this case the phenomenon should be classified as "evaporation"
In fact (from Cambridge dictionary): fumigation: the use of poisonous gas to remove harmful insects, bacteria, disease, etc. from somewhere or something.
Line 155: atop??
Lines 156-57 (figure 1): fig. 1C is not clear: where are the rows? (I mean rows 1, 2 and 3? and the circles? and the control ones? please redraw in a clearer way
Lines 185-198: control plot is not mentioned; 195-196: why you write this sentence? I think is better to erase since is a repetition
208-209: from a statistical point of view it is certainly not the best thing………; again, the control? are you sure that the control (where is?) was not affected by the evaporation?
Lines 310-312: so, you affirm that different environmental conditions during the three replicates did not influence the performance of EOs? whereas the phytotoxicity was different? In which way could you explain these differences?
Lines 374-75: if the putative effects of EOs are jointed to AChE is not a good news! we would expect that after two-three treatments with EOs the appearance of resistance...aphids are so quick in this field!!
Lines 421-22: this is in contrast with your previous sentence!! see rows 310-312....
Lines 451-57: s not a good example: Prev-Am is a fungicide-insecticide that acts only by cuticular contact; thus N. tenuis could be not affected being extremely mobile; further it could be interesting to stress the possible action of EOs on predators and parasitoids normally associated with this pest, mainly hoverflies larvae, micro_Hymenoptera and ladybirds. Please, rewrite this sentence in the light of the possible effects of EOs on these insects.
Author Response
Dear Reviewer 1, Thanks for your time and relevant comments. Please find our answers below
Lines 28-30: erase this sentence; is not appropriate for the abstract
Corrected and moved in the conclusion part
Line 101: Material instead of materiel
Corrected
Lines 117-118: rewrite the sentence that is not so clear
We removed the sentence which was indeed not so clear
Lines 122-123: what kind of formulation?
No particular formulation, we used pure Eos in this study as specified L143
Line 144 onwards: are you sure that you could use the term fumigation in this experiment? I think that in this case the phenomenon should be classified as "evaporation"
In fact (from Cambridge dictionary): fumigation: the use of poisonous gas to remove harmful insects, bacteria, disease, etc. from somewhere or something.
In their review « Essential oils as active ingredients of botanical insecticides against aphids » Ikbal & Pavela 2019 compare the efficacy of Eos using laboratory methods against aphid species in contact and fumigation tests. We re-used this term by opposition of contact methods in order to fit well with this closely-related review.
Line 155: atop??
Corrected and replaced by above
Lines 156-57 (figure 1): fig. 1C is not clear: where are the rows? (I mean rows 1, 2 and 3? and the circles? and the control ones? please redraw in a clearer way
We clarified Fig 1C
Lines 185-198: control plot is not mentioned;
The control was mentioned L145 for all set-up (three scales). We repeat it L197 to avoid any confusion.
195-196: why you write this sentence? I think is better to erase since is a repetition
In the first sentence, we specified which dose was used, adapted to the air volume of the climatic chamber. In the second sentence, we specified this dose was divided between all filter papers that were introduced in the climatic chamber. We rephrased in order to make it clearer.
208-209: from a statistical point of view it is certainly not the best thing………; again, the control? are you sure that the control (where is?) was not affected by the evaporation?
The confusion might come from a misunderstanding here. For each trial (3 trials = 3 time periods), each greenhouse compartment was submitted to one treatment: Control = water or Eos green anise or Eos fennel i.e. all treatments were applied and compared for all trials. Furthermore, despite the seasonal variability, we still have a clear treatment effect. The set-up was clarified in the text and in Fig 1C as well. The word “replicate” was replaced by “trial”. We clarified on Fig. 5 as well.
Lines 310-312: so, you affirm that different environmental conditions during the three replicates did not influence the performance of EOs? whereas the phytotoxicity was different? In which way could you explain these differences?
The misunderstanding here could be the same one than in the previous question! Each treatment (control EO anise and Eos fennel) was applied at each trial = replicate time. So, when we detect significant differences between treatments, it includes the seasonal variability. In the discussion we both discussed the differences between treatments i.e. our results on the efficiency of Eos as biopesticides (4.1) but also the potential variability linked to environmental conditions (4.3.).
Lines 374-75: if the putative effects of EOs are jointed to AChE is not a good news! we would expect that after two-three treatments with EOs the appearance of resistance...aphids are so quick in this field!!
In the present study, aphids were in contact only once with the Eos. Our experimental set-up is not appropriate to check for resistance (was not the goal anyway for this first study), but we agree this question is an important issue. We add it in the conclusion as perspective for coming projects.
Lines 421-22: this is in contrast with your previous sentence!! see rows 310-312....
Again, we are confronted here to the same misunderstanding: the difference between treatments (Control vs Eos green anise or fennel) was variable, although always significant, and this variability was linked to climatic conditions. Our conclusion was that Eos are efficient but more or less regarding climatic conditions. By clarify the experimental set-up in the greenhouse, we think we will avoid the misunderstanding in the discussion part.
Lines 451-57: s not a good example: Prev-Am is a fungicide-insecticide that acts only by cuticular contact; thus N. tenuis could be not affected being extremely mobile; further it could be interesting to stress the possible action of EOs on predators and parasitoids normally associated with this pest, mainly hoverflies larvae, micro_Hymenoptera and ladybirds. Please, rewrite this sentence in the light of the possible effects of EOs on these insects.
We agree with reviewer 1 that the example we choose was not the most relevant one. Still, We keep it, but shorten, in order to illustrate a combination between EO-based biopesticides and biocontrol. We question then the potential combination between EO fumigation methods and biocontrol, based on non-published results.
Reviewer 2 Report
- There are some language issues and here are some examples- In Line 15 "Their regulation" probably best to say their control.
- Line 18 "three increasing scales" better say "three different scales."
- In line 22 after the word air write" respectively."
- Line 360 word "we" probably they meant to write when.
- My second point is about the methods used. The methods such as EO in open petri dishes will not work at all in the field and therefore, the authors should have talked and considered using spray method or how to make it stick to plants and be effective on the plants. None of these have been discussed.
- Third, the authors did not mention whether the smell of EO have effect on human or not or whether the concentrations used in greenhouse was tolerable or not.
Author Response
Dear Reviewer 2, Thanks for your time and relevant comments. Please find our answers below
- There are some language issues and here are some examples- In Line 15 "Their regulation" probably best to say their control.
Corrected + The MS was proof-read by the office of International Scientific Visibility of the University UCA
- Line 18 "three increasing scales" better say "three different scales."
Corrected
- In line 22 after the word air write" respectively."
Corrected
- Line 360 word "we" probably they meant to write when.
Corrected
- My second point is about the methods used. The methods such as EO in open petri dishes will not work at all in the field and therefore, the authors should have talked and considered using spray method or how to make it stick to plants and be effective on the plants. None of these have been discussed.
We agree with Reviewer 2 that the fumigation method is not an appropriate one for open field. However, our greenhouse-scale experiment show that Eos fumigation might be used and be efficient in such situation. This is why we built our discussion at the agronomic greenhouse setting scale (L336 and 345, 441).
- Third, the authors did not mention whether the smell of EO have effect on human or not or whether the concentrations used in greenhouse was tolerable or not.
In case of direct contact, like for most of the cosmetic products, the presence and concentration of a relatively potent allergen is a major factor in allergic contact dermatitis. Photosensitization may also occur when a phototoxin in the essential oil is applied to the skin in the presence of sunlight or ultraviolet light. A list of 26 possible allergenic fragrances have been defined (https://doi.org/10.3390/molecules26030666), of which 18 can be found in essential oils. For this reason, they must be declared on the packaging or in the information leaflet if the concentration of these allergenic fragrances is higher than the permissible concentration (different according to the cosmetic formulations).
On the other hand, a recent report from the ANSES (Agence nationale de sécurité sanitaire de l'alimentation), says that the scientific literature available on sprays and diffusers based on essential oils is insufficient to draw any conclusions about their toxicity on human health (https://www.anses.fr/fr/system/files/AIR2018SA0145Ra.pdf). The publications analysed and the elements extracted from them provide, however, signals that call for vigilance.
In our study, and as mentioned in the experimental set-up, compartments were locked for 24h after Eos application then ventilated for 24h (automatic windows). Then, experimenters were exposed to Eos fumigation only during the application phase in an open ventilated greenhouse but never during the closing phase.
Reviewer 3 Report
In the ms the authors present the results of three scales evaluation of EOs against potato aphid. Due to some potential and actual side effects of essential oils, the authors performed experiment in three levels. They used suitable methods for obtaining the results and suitable analysis for comparing the results between treatments. After carefully reading the ms I suggest to improve the ms in the follwoing places:
line 16: replace "Human" with "human"
line 17: replace "Botanical biopesticides" with "Botanical pesticides"
line 61: I suggest to add some more references, for example: ROJHT et al., 2012. Chemical analysis of three herbal extracts and observation of their activity against adults of Acanthoscelides obtectus and Leptinotarsa decemlineata using a video tracking system. Journal of plant diseases and protection, 119, 2: 59-67.
line 73: when the organism is first mentioned in the text it should be written with full Latin name, for example: Foeniculum vulgare Mill. This should be taken into account also with other organisms (plants, insects...) in the paper!
lines 179-184: Is this a standard scale? If yes, you should mention the reference. If no, you should mention that the scale was made by the authors.
Author Response
Dear Reviewer 3, Thanks for your time and relevant comments. Please find our answers below
line 16: replace "Human" with "human"
corrected
line 17: replace "Botanical biopesticides" with "Botanical pesticides"
corrected
line 61: I suggest to add some more references, for example: ROJHT et al., 2012. Chemical analysis of three herbal extracts and observation of their activity against adults of Acanthoscelides obtectus and Leptinotarsa decemlineata using a video tracking system. Journal of plant diseases and protection, 119, 2: 59-67.
We thank Reviewer 3 for the good suggestion and we add a recent review on the topic which includes many experimental references. However we will not quoted the reference suggested by Reviewer 3 as it focus on plant extract in ethanol when our sentence focus on essential oils mode of action.
line 73: when the organism is first mentioned in the text it should be written with full Latin name, for example: Foeniculum vulgare Mill. This should be taken into account also with other organisms (plants, insects...) in the paper!
We agree the full latin names must be mentioned, particularly. We mention them in the Mat& Meth part when we described each studied organism.
lines 179-184: Is this a standard scale? If yes, you should mention the reference. If no, you should mention that the scale was made by the authors.
The scale was made by the authors, we specify in the text.